# UNIFYING USER PREFERENCES AND CRITIC OPINIONS: A MULTI-VIEW CROSS-DOMAIN ITEM-SHARING RECOMMENDER SYSTEM

## ABSTRACT

Traditional cross-domain recommender systems often assume user overlap and similar user behavior across domains. However, these presumptions may not always hold true in real-world situations. In this paper, we explore an less explored but practical scenario: cross-domain recommendation with distinct user groups, sharing only item-specific data. Specifically, we consider user and critic review scenarios. Critic reviews, typically from professional media outlets, provide expert and objective perspectives, while user reviews offer personalized insights based on individual experiences. The challenge lies in leveraging critic expertise to enhance personalized user recommendations without sharing user data. To tackle this, we propose a Multi-View Cross-domain Item-sharing Recommendation (MCIR) framework that synergizes user preferences with critic opinions. We develop separate embedding networks for users and critics. The user-rating network leverage a variational autoencoder to capture user scoring embeddings, while the user-comment network use pretrained text embeddings to obtain user commentary embeddings. In contrast, critic network utilize multi-task learning to derive insights from critic ratings and reviews. Further, we use Graph Convolutional Network layers to gather neighborhood information from the user-item-critic graph, and implement an attentive integration mechanism and cross-view contrastive learning mechanism to align embeddings across different views. Real-world dataset experiments validate the effectiveness of the proposed MCIR framework, demonstrating its superiority over many state-of-the-art methods.

## 1 INTRODUCTION

Cross-domain recommendation systems have attracted considerable attention recently due to their ability to mitigate issues such as data sparsity in recommender systems by leveraging auxiliary information from associated domains (Zhu et al., 2020; Hu et al., 2018; Kang et al., 2019; Chen et al., 2022). Traditional cross-domain recommenders often presume an overlap of users and similar user types across domains (Singh & Gordon, 2008; Hu et al., 2018; Yan et al., 2019), and they typically share user-related information across domains to enhance recommendation performance. However, real-world scenarios may present entirely different user types across auxiliary and target domains, and sharing user-centric information might not be practical or permissible due to privacy issues or operational constraints. Addressing this, our study delves into a less charted yet practical area within cross-domain recommendation systems where solely item-related information is shared between disparate user types in the auxiliary and target domains. Specifically, we concentrate on the user and critic review scenarios. Noticeably, without loss of generality, our framework can be easily applied in analogous item-sharing situations.

The fast growth of review websites like Metacritic (met) and ROTTEN TOMATOES (rot) offers a space where users can not only share their perspectives but also gain from the expert evaluations of critics. Critic reviews, predominantly sourced from authoritative media institutions, can give professional and objective insights, serving as an invaluable resource for decision-making users. In contrast, user reviews often echo personalized, experience-centric perspectives. The substantial impact of critic reviews on user preferences is well documented in existing marketing research (Basuroy et al., 2003; Tsao, 2014). Researchers also underscore the divergent nature of critiques offered by critics and users (Santos et al., 2019; Dillio, 2013; Parikh et al., 2017), such as "Experts Write More Complex and Detached Reviews, While Amateurs More Accessible and Emotional Ones". This

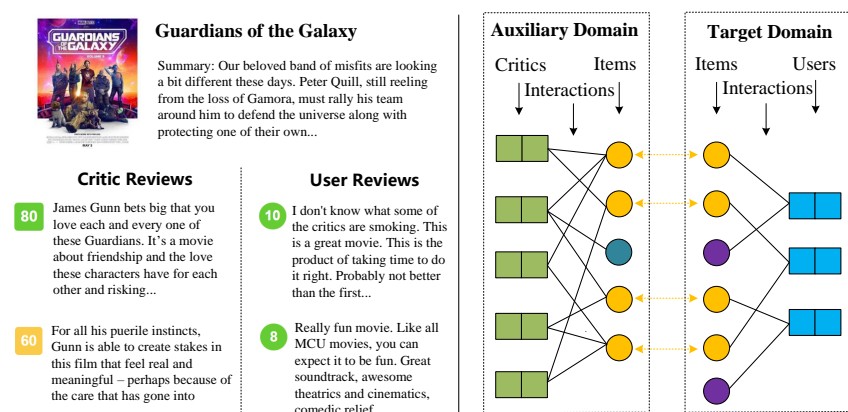

Figure 1: An example of the user and critic review scenario (left) and the illustration of cross-domain recommendation without sharing users (right).

distinction accentuates the importance of integrating expert knowledge from the critic domain to enhance the quality and reliability of recommendations in the user domain.

However, transferring critic information into the user domain presents several challenges. First, user comments can be diverse and may lack correlation with item properties (Parikh et al., 2017), making it crucial to learn comprehensive user preferences from both rating and comment views. Second, as illustrated in Figure 1, critic reviews can also greatly differ from each other, making it necessary to identify the most influential critiques for users. Third, there are no direct links between users and critics, making it necessary to use items as a bridge to capture consistency. Hence, leveraging critic domain information to enhance personalized user recommendations becomes a complex yet rewarding task (Gao et al., 2019; 2021).

To address these challenges, we propose a novel Multi-view Cross-domain Item-sharing Recommendation (MCIR) framework that effectively synthesizes user preferences and critic opinions. We initially design unique embedding networks tailored to users and critics for learning multi-view information. The user-rating network employs a variational autoencoder to capture user scoring embeddings, while the user-comment network utilizes pretrained text embeddings to obtain user commentary embeddings. Conversely, the critic network employs multi-task learning to derive insights from critic ratings and comments synchronously. Based on the multi-view representations, we devise an attentive integration mechanism to obtain comprehensive item representations. Further, we extract detailed neighborhood information from the user-item-critic graph and propose a cross-view contrastive learning method to harmonize embeddings across different views. Extensive experiments on real-world datasets demonstrate that our proposed MCIR framework outperforms state-of-the-art methods, effectively addressing the challenges of cross-domain item-sharing recommendations.

## 2 RELATED WORKS

Cross-domain recommendation (CDR) is a widely-used technique to counter challenges like data sparsity by incorporating data from auxiliary domains. CDR mainly encompasses collaborative and content-based approaches (Mirbakhsh & Ling, 2015; Gao et al., 2021).

On one hand, collaborative CDR methods draw on interaction data across domains. For example, Collective Matrix Factorization (CMF) (Singh & Gordon, 2008) is a classic CDR approach, assuming a global user factor matrix across all domains while factorizing multiple domain matrices. Differently, Man et al. (2017) used a multi-layer perceptron to capture nonlinear mapping across domains. Similarly, DCDCSR (Zhu et al., 2020) combined user latent vectors and learns a mapping function between target domains. CoNet (Hu et al., 2018) used cross-connections between neural networks to transfer and consolidate knowledge. SSCDR (Kang et al., 2019) merged collaborative filtering with sparse subspace clustering to enhance recommendation systems by aligning latent subspaces. DeepAPF (Yan et al., 2019) modeled user-video interactions by capturing cross-site and site-specific interests with an attentional network. BiTGCF (Liu et al., 2020) used a feature propagation layer for high-order connectivity within a domain's user-item graph, enabling knowledge transfer. CAT-ART (Li et al., 2023) and COAST (Zhao et al., 2023) improved performance across domains through representation learning, embedding transfer, and aligning user interests. Besides, for addressing privacy, PriCDR (Chen et al., 2022) employed a privacy-preserving CDR model.

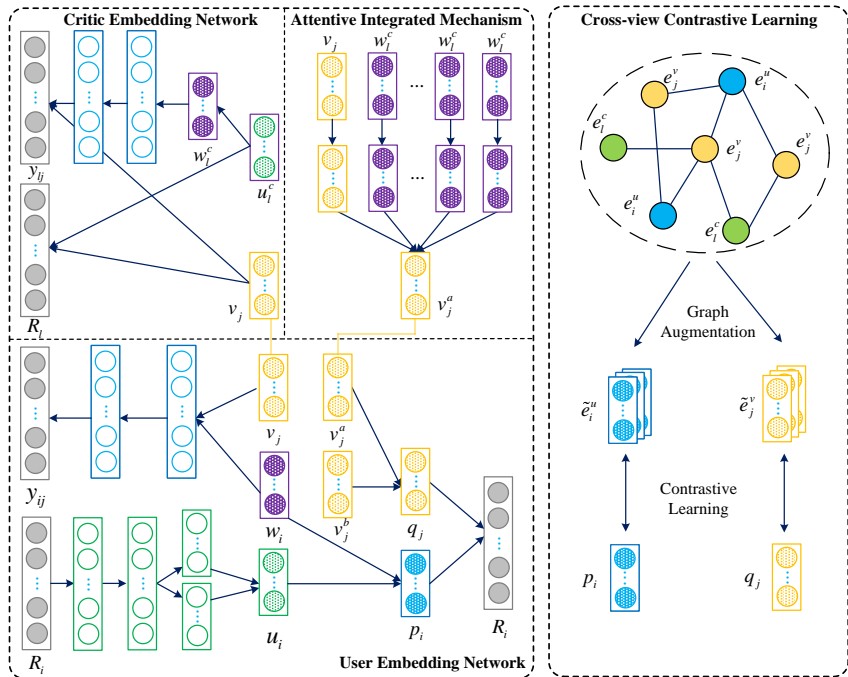

Figure 2: The network architectures of the MCIR Framework.

On the other hand, content-based CDR methods utilize user or item attributes from auxiliary domains. For instance, LFM (Agarwal et al., 2011) used multi-modal user profiles, and CKE (Zhang et al., 2016) enhanced item embeddings with textual, structural, and visual knowledge. CATN (Zhao et al., 2020) modeled user preference transfer from reviews, while CCDR (Xie et al., 2022) addressed popularity bias through a diversified preference contrastive learning.

Although Cross-Domain Recommendation (CDR) methods have achieved notable successes in academic literature, particularly for user-sharing scenarios, there remains a gap in addressing contexts where only items are shared across varied user types. Recently, Gao et al. (2019; 2021) performed cross-domain recommendations without sharing user-sided data, but these efforts persisted in assuming uniform user types across domains. Contrary to preceding studies, our work uniquely concentrates on scenarios where the auxiliary and target domains are characterized by distinct user types, with only items being a consistent shared entity across the domains.

## 3 THE MCIR FRAMEWORK

In this section, we delve into the technical specifics of our proposed Multi-View Cross-domain Item-sharing Recommendation (MCIR) framework. Initially, we present the relevant notations and outline our framework. Subsequently, we provide a detailed explanation of the multi-view learning process in the MCIR framework. Finally, we combine the different views to generate recommendations.

### 3.1 PROBLEM DEFINITION

In this study, we deal with two distinct domains of data. Let's assume there are $N_u$ users, $N_v$ items, and $N_w$ critics. In the user domain, we define the rating matrix $R \in \mathbb{R}^{N_u \times N_v}$ as the composition of historical ratings represented by real numbers, with missing entries denoted by 0. Let $R_i$ represent the rating records of the $i$-th user across all items. Besides, we use $Y_{ij}$ to represent the comment text by user $i$ on item $j$ and $Y_i = \{Y_{ij} | I(Y_{ij}) = 1\}$ to denote the set of user $i$'s comments. In the critic domain, we similarly use $R_{lj}^c$ and $Y_{lj}^c$ to represent the rating and comment of critic $l$ on item $j$, respectively. Note that in the following text, variables marked with a superscript c (e.g., $Y_{lj}^c$ and $R_{lj}^c$) all denote variables within the critic domain. Then the problem can then be defined as:

**Definition 1.** *(Cross-domain Item-sharing Problem.) Consider the auxiliary domain data (comprising rating matrix $R^c$ and comment records $Y^c$) and the target domain data ($R$ and $Y$). The objective is to make item recommendations to target users by harnessing information from both the target and auxiliary domains. This scenario is characterized by the exclusive sharing of item-relevant data, with no user overlap between the target and auxiliary domains.*

## 3.2 Framework Overview

As depicted in Figure 2, our MCIR framework adeptly performs multi-view representation learning to precisely model cross-domain interactions. Initially, the input $R_i$ is encoded from a user-rating perspective, producing latent rating vectors $u_i$ that embody user preferences from their rating histories. Following this, comment records $Y_i$ are used to ascertain latent user-comment vectors $w_i$. These two views are subsequently merged to formulate a unified latent user vector. In the critic domain, a multi-task network is utilized to simultaneously discern both the critic-rating vector $u_i^c$ and the critic-comment vector $w_l^c$. Concerning items, latent item-text vectors are derived from item summary texts, and latent item-rating vectors are gleaned from both user and critic domains. Our framework further leverages the user-item-critic graph to unearth latent neighborhood vectors, enhancing the capture of user preferences and item attributes from a neighborhood perspective. To ensure alignment across all views, a cross-view contrastive learning mechanism is proposed. Concluding the framework's operation, the user decoder network seamlessly performs the rating prediction, facilitating robust and informed item recommendations.

## 3.3 Methodology

To tackle the challenge of cross-domain recommendation in the absence of shared users, we devise a novel solution framework, termed MCIR, which harnesses multi-view knowledge to adeptly discern user preferences and item properties across various domains.

**User-Rating Embedding Network.** Given the diverse nature of user comments and their potential limited correlation with item property evaluations (Santos et al., 2019; Dillio, 2013; Parikh et al., 2017), we opt for independent learning of two distinct types of latent embeddings from both user-rating and user-comment perspectives, respectively. Initially, from the user-rating view and inspired by the famous matrix factorization models for collaborative filtering (Mnih & Salakhutdinov, 2008), our goal is to decompose the rating matrix into two latent representations $U \in \mathbb{R}^{d \times N_u}$ and $V \in \mathbb{R}^{d \times N_v}$ in a shared low-dimensional space of dimension $d$. Hence we can use $u_i \in \mathbb{R}^d$ and $v_j \in \mathbb{R}^d$ to represent the latent factors of user $i$ and item $j$ from the rating perspective.

To guarantee robust and efficient embedding learning process, we utilize the Variational Autoencoder (VAE) due to its well-documented proficiency in generating accurate and robust recommendations (Liang et al., 2018; Ma et al., 2019). VAE can generate more reliable and effective embeddings rather than common Autoencoders (AE) in recommender systems (Khawar et al., 2020). Accordingly, the encoder network $f_\psi(\cdot)$ of VAE, named User-Rating Embedding Network, is structured as multilayer perceptrons (MLPs) with $T$ layers. The user-rating vector $u_i$ is thus expressed as a multivariate Gaussian variable with mean $\mu$ and covariance $\Sigma$, computed as follows:

$$
\begin{aligned}
h_1 &= g(W_1 f_{drop}(R_i) + b_1), \\
h_t &= g(W_t h_{t-1} + b_t), \quad t \in [2, 3, ..., T], \\
\mu_i &= W_T h_T + b_T, \quad diag\{\Sigma_i\} = W_T' h_T + b_T',
\end{aligned}
\tag{1}
$$

where $diag\{\Sigma_i\}$ denotes the diagonal elements of the matrix $\Sigma$, with all other elements set to 0. $g(\cdot)$ represents the activation function, and $f_{drop}(\cdot)$ signifies the drop-out layer. Employing a drop-out strategy on $R_i$ markedly reduces the overfitting problem, leading to more robust representations.

The distribution $p(u_i|R_i)$ is then derived as $p(u_i|R_i) \sim \mathcal{N}(\mu_i, \Sigma_i)$. By utilizing the reparameterization trick (Rezende et al., 2014), the sampling on variable $u_i$ during the gradient backpropagation process is avoided, and $u_i$ is obtained via $u_i = \mu_i + \epsilon \Sigma_i$, where $\epsilon \sim \mathcal{N}(0, I)$ is the multivariate Gaussian noise. It's important to note that unobserved entries are represented as 0 in the rating matrix. Hence, using the entire record vector $R_i$ as network input enables us to learn not only user evaluation scores but also interactive preferences for item selection.

**User-Comment Embedding Network.** Upon transposing the rating information into a latent dimension, we then need to extract representations from the comment view. Specifically, the latent user-comment vector $w_i \in \mathbb{R}^d$ is initialized randomly and updated during the training process.

First, we employ the famous pretrained Sentence-BERT (Reimers & Gurevych, 2019) to extract paragraph embeddings $y_{ij} \in \mathbb{R}^b$ for each comment text $Y_{ij}$. The framework of Sentence-BERT, trained to discern semantically similar and dissimilar sentence pairs, aids in extracting consistent latent user-comment vector $w_i$ for users with analogous review patterns. Simultaneously, acknowledging the item summary's encapsulation of crucial properties, Sentence-BERT is utilized to yield the pretrained embedding $s_j \in \mathbb{R}^b$ for the summary text of item $j$. Both $y_{ij}$ and $s_j$ remain fixed in subsequent processes. For further gleaning item property information that is valuable for recommendations,

a linear layer $f_s(\cdot)$ is introduced to convert the original item summary embedding into the latent item-summary vector $v_j = f_s(s_j) \in \mathbb{R}^d$.

Since user comments encapsulate both user preferences and item characteristics, we merge the latent user-comment vector $w_i$ and item-summary vector $v_j$ to reconstruct the pretrained embedding $y_{ij}$ for user $i$'s review on item $j$. A 2-layer MLP network $f_z(\cdot)$ is deployed to produce the reconstructed comment vector $z_{ij} \in \mathbb{R}^b$ for review $Y_{ij}$ as follows, where $\cdot||\cdot$ denotes concatenation.

$$z_{ij} = f_z(w_i||v_j). \tag{2}$$

The reconstruction aspires $z_{ij}$ to closely mirror $y_{ij}$. Discrepancies between $z_{ij}$ and $y_{ij}$ are quantified using Mean Square Error (MSE) loss:

$$\mathcal{L}_{MSE} = \Sigma_i \Sigma_j I(Y_{ij})(\|y_{ij} - z_{ij}\|^2 + \lambda\|w_i\|^2 + \lambda\|v_j\|^2), \tag{3}$$

where $I(Y_{ij})$ is the identity function which equals 1 if user $i$ has reviewed item $j$, otherwise 0. The inclusion of an L2 regularization term for latent vectors with the weight hyperparameter $\lambda$ enhances the model's robustness. This loss function aids in aligning the reconstructed comment vector closely with the original, ensuring the infusion of efficient text information into $w_i$ and $v_j$.

**Critic Embedding Network.** In this subsection, we aim to learn latent critic representations. Given the unique nature of critics—marked by professional insights and objective commentaries—two distinct features set them apart from ordinary users. On one hand, critics, often affiliated with credible media entities, lack the liberty to select items for review based on personal preferences. Their ratings mainly echo their assessments of the items. On the other hand, a significant correlation exists between critics' scores and their explanatory comments (Santos et al., 2019; Dillio, 2013). Therefore, here we employ a multi-task way to concurrently learn the latent critic-rating and critic-comment vectors.

For the critic rating prediction task, we define $u_l^c \in \mathbb{R}^d$ as the latent variable for critic $l$. Following the matrix factorization model, critic-item ratings $\hat{R}_{lj}^c$ are predicted through the inner product of the latent critic-rating vector $w_l^c$ and the item vector $v_j$, expressed as $\hat{R}_{lj}^c = (u_l^c)^T v_j$ (Koren et al., 2009; Xue et al., 2017). Noticed that here we employ the previously outlined latent item-summary vector $v_j$ (Equation 2) to maintain coherence across the auxiliary and target domains.

Considering the robust association between critics' ratings and comments, a transformation of the latent critic vector $u_l^c$ into the critic-comment vector $w_l^c = f_w(u_l^c)$ is performed using a single-layer MLP network $f_w(\cdot)$. Again Sentence-BERT is used to obtain the pretrained review embedding $y_{lj}^c$ for critic review $Y_{lj}^c$. Similarly to the User-Comment Embedding Network, a 2-layer MLP network $f_c(\cdot)$ processes the merged vector $w_l^c||v_j$ to recreate the original critic comment embedding $y_{lj}^c$. This leads to the following mapping function for the latent critic and item-summary representations:

$$z_{lj}^c = f_c(w_l^c||v_j). \tag{4}$$

To quantify the variations from both rating and comment perspectives, a multi-task loss is employed:

$$\mathcal{L}_{Multi} = \Sigma_l \Sigma_j I(Y_{lj}^c)(\|\hat{R}_{lj}^c - R_{lj}^c\|^2 + \|y_{lj}^c - z_{lj}^c\|^2 + \lambda\|w_l^c\|^2 + \lambda\|v_j\|^2). \tag{5}$$

**Attentive Integrated Mechanism.** Given that only the items are shared between the critic and user domains, it is crucial to leverage items as a conduit for message exchange. In this subsection, an attentive integrated mechanism is proposed, aimed at generating the attentive item vector $v_j^a$. This vector seamlessly combines both the item-summary information and the critics' commentary information, thereby enhancing the recommendations within the user domain.

Given the potential diversity in critics' perspectives on different aspects of an item, leading to substantial disagreements in their reviews, a thorough understanding of item characteristics is essential. This understanding aids in the generation of attentive item embeddings from the wide range of critic comments. To achieve this integration, an attention layer is introduced to merge the item-summary vector and pertinent latent critic vectors. Let $L_j$ be the set containing the indices of critics who have reviewed item $j$. For the convenience of establishing formulas, let $w_0^c = v_j$ and include index 0 in the set $L_j$. Then the attentive integrated mechanism is articulated as:

$$
\begin{aligned}
\alpha_{jl} &= \frac{exp((W_{key}v_j)^T(W_{query}w_l^c))}{\Sigma_{l' \in L_j}, exp((W_{key}v_j)^T(W_{query}w_{l'}^c))}, l \in L_j, \\
v_j^a &= \Sigma_{l \in L_j} \alpha_{jl}(W_{value}w_l^c). 
\end{aligned}
\tag{6}
$$

In Equation 6, $\alpha_{jl}$ denotes the attention weights, determined by the compatibility between the item-summary vector $v_j$ and critic vectors $w_l^c$. The symbols $W_{\text{key}}$, $W_{\text{query}}$ and $W_{\text{value}} \in \mathbb{R}^{d \times d}$ represent the learnable weight matrices for the key, query, and value in the attention mechanism, respectively. The final attentive item embedding $v_j^a$ is procured as the weighted sum of the critic-comment vectors, with the attention scores $\alpha_{jl}$ as weights. Practically, for large $|L_j|$, a subset of the critics is sampled in each batch for training, employing a random sampling strategy. This strategy is not only efficient but also serves as a preventive measure against over-fitting. Through this attentive integration mechanism, diverse critic information is effectively amalgamated, encompassing both the item-summary information and critics' opinions, thus offering a more enriched and comprehensive item representation for recommendations in user domain.

**User Decoder Network.** The next phase is to decode the latent user and item vectors obtained from multi views back into the original user-item rating space. Specifically, we first combine the latent user-rating vector and user-comment vector to produce the final latent user representation $p_i$. As for the items, since the textual information may not include all the useful information for recommendations, we add a learnable variable $v_j^b \in \mathbb{R}^d$ to the attentive vector $v_j^a$ to obtain the final item representation $q_j$. $v_j^b$ is initialized randomly and updated throughout the training process.

$$p_i = u_i + w_i, \quad q_j = v_j^a + v_j^b. \tag{7}$$

In Equation 7, the addition operation is preferred over concatenation to maintain a consistent dimension of the latent vector. This approach also lays the foundation for the addition of more potential views. The user decoder network can hence be defined by:

$$\hat{R}_{ij} = p_i^T q_j. \tag{8}$$

The VAE loss can be formulated as:

$$\mathcal{L}_{VAE} = -\Sigma_i \left( \mathbb{E}[\log p(R_i|p_i)] + \mathbb{KL}(p(u_i|R_i)|\mathcal{N}(0, I)) \right). \tag{9}$$

The first component in Equation 9 is the reconstruction loss. Following the method in VAE, Monte Carlo sampling aids in estimating expected values. However, given that $R_i$ is not strictly binary, cross-entropy loss application is not straightforward for estimating the term $\mathbb{E}[\log p(R_i|p_i)]$. As an alternative, inspired by Xue et al. (2017), a softmax layer $\hat{R}_i' = softmax(\hat{R}_i)$ is first applied, and a novel reconstruction loss is defined as:

$$\log p(R_i|p_i) = -\Sigma_j \frac{R_{ij}}{max(R_i)} \log \hat{R}_{ij}'. \tag{10}$$

Compared to the loss in (Xue et al., 2017), our proposed loss in Equation 10 computes the probability across the entire rating vector $R_i$, rather than focusing on a specific item.

**Graph Embedding Network and Cross-view Contrastive Learning.** In addition to rating and comment perspectives, the neighborhood perspective can also offer important insights into user preferences, item characteristics, and critic opinions (He et al., 2020; Mao et al., 2021). In order to capture these relationships and uphold consistency across domains, a user-item-critic graph is formulated to learn neighborhood embeddings. The nodes of this graph include all users, items, and critics, and edges denote positive interactions. We filter these positive interactions by selecting ratings above a certain threshold. The neighborhood information enables the integration of user and critic preferences within a unified view, with items serving as the connecting bridge.

Specifically, the efficient and widely recognized LightGCN (He et al., 2020) architecture is employed to obtain latent graph vectors $e_i^u$, $e_l^c$, and $e_j^v$ for user $i$, critic $l$, and item $j$ respectively. LightGCN eliminates feature transformations and non-linear activation functions. Let $\mathcal{N}_i$, $\mathcal{N}_l$, and $\mathcal{N}_j$ signify the neighborhood node sets of node $i$, $l$, and $j$ respectively. The message passing layer is denoted as:

$$e_{i,(k+1)}^u = \Sigma_{j \in \mathcal{N}_i} \frac{e_{j,k}^v}{\sqrt{|\mathcal{N}_i|}\sqrt{|\mathcal{N}_j|}}, \quad e_{l,(k+1)}^c = \Sigma_{j \in \mathcal{N}_l} \frac{e_{j,k}^v}{\sqrt{|\mathcal{N}_l|}\sqrt{|\mathcal{N}_j|}},$$

$$e_{j,(k+1)}^u = \Sigma_{i \in \mathcal{N}_i} \frac{e_{i,k}^u}{\sqrt{|\mathcal{N}_i|}\sqrt{|\mathcal{N}_j|}} + \Sigma_{l \in \mathcal{N}_l} \frac{e_{l,k}^c}{\sqrt{|\mathcal{N}_l|}\sqrt{|\mathcal{N}_j|}}, \tag{11}$$

where $k$ represents the ordinal number of a GCN layer and the final latent graph vectors are computed as the average of all the $K$ layer's embeddings to prevent the over-smoothing problem.

For the neighborhood view, the rating prediction function is $\hat{R}_{ij}^g = (e_i^u)^T e_j^v$ and $\hat{R}_{il}^g = (e_l^c)^T e_j^v$. Hence, the GCN loss can be given similar to Equation 10:

$$\mathcal{L}_{Graph} = -\Sigma_i\Sigma_j \frac{R_{ij}}{max(R_i)} \log \hat{R}_{lj}^g - \Sigma_l\Sigma_j \frac{R_{lj}}{max(R_l)} \log \hat{R}_{lj}^g + \lambda(\Sigma_i\|e_i^u\|^2 + \Sigma_l\|e_l^c\|^2 + \Sigma_j\|e_j^v\|^2). \tag{12}$$

The GCN layer can effectively capture the neighborhood information, thereby enriching the understanding of user preferences and critic opinions. However, the user-item-critic graph mainly comprises positive implicit interactions. A direct addition of graph vectors to $p_i$ or $q_j$ hinders rather than enhances performance. To navigate this challenge, a cross-view contrastive learning mechanism is proposed, ensuring alignment of vectors across varying views without the direct addition or concatenation of representations.

Contrastive learning facilitates comparisons between diverse augmented samples and have been proven effective in recommender systems (Wu et al., 2021). Using graph sampling techniques, augmented latent graph vectors $\widetilde{e}_i^u$ and $\widetilde{e}_j^v$ are generated. Unlike traditional contrastive learning methods like InfoNCE (Gutmann & Hyvärinen, 2010), which consider two samples from the same view as positive pairs, this approach treats the unified vector $p_i$ and the augmented graph vector $\widetilde{e}_i^u$ as a positive pair. $p_i$ and other augmented graph vectors $\widetilde{e}_{i'}^u, i' \neq i$ are treated as negative pairs. In each batch, we randomly sample some negative pairs. The same approach is applied to items. Formally, we can maximize positive pair agreement and minimize negative pair agreement as follows:

$$\mathcal{L}_{CL} = -\Sigma_i \log \frac{\exp(\cos(p_i, \widetilde{e}_i^u)/\tau)}{\exp(\cos(p_i, \widetilde{e}_i^u)/\tau) + \Sigma_{i'} \exp(\cos(p_i, \widetilde{e}_{i'}^u)/\tau)}$$
$$-\Sigma_j \log \frac{\exp(\cos(q_j, \widetilde{e}_j^v)/\tau)}{\exp(\cos(q_j, \widetilde{e}_j^v)/\tau) + \Sigma_{j'} \exp(\cos(q_j, \widetilde{e}_{j'}^v)/\tau)}, \tag{13}$$

Here, $\cos(\cdot)$ is the cosine similarity function, and $\tau_c$ is the temperature parameter.

Finally, the unified loss within the MCIR framework is defined as:

$$\mathcal{L} = \mathcal{L}_{VAE} + \eta_1 \mathcal{L}_{MSE} + \eta_2 \mathcal{L}_{Multi} + \eta_3 \mathcal{L}_{Graph} + \eta_4 \mathcal{L}_{CL}. \tag{14}$$

Noticeably, while our primary focus is on user and critic review scenarios, MCIR can actually be effortlessly adapted to similar item-sharing contexts with minimal modifications for varying features.

## 4 EXPERIMENTS

In this section, we begin by detailing the datasets, evaluation protocols, baseline methods, and experimental settings. Then, we report the recommendation performance results of our proposed MCIR models compared to the state-of-the-art baselines. Further, we conduct ablation studies to validate the efficacy of each MCIR component. Discussion on the impact of the cross-view contrastive learning mechanism and the influence of various hyper-parameters on the performance is included. Finally, we present some case studies in the Appendix to show the explanatory capabilities of MCIR.

### 4.1 EXPERIMENTAL SETTINGS

**Datasets.** The datasets used in the experiments were collected from Metacritic [1]. We collect the user and critic reviews as well as ratings for games, movies, and musics up to December 2022 to form three datasets, i.e., *MC-Game*, *MC-Movie*, and *MC-Music*. [2] In the *MC-Game*, *MC-Movie*, and *MC-Music* datasets, user ratings are expressed as 10-stars, whereas critic scores utilize a percentage system. To facilitate comparison, we normalize both user and critic scores to fall within the $[0, 1]$ range. For validation, following (Wu et al., 2018), we adopted the data preprocessing to differentiate the positive and negative feedback depending on whether the ratings are not less than 0.7. *MC-Game* contains 18,622 users, 522 critics, and 16,713 items with 505,964 user reviews and 242,764 critic reviews. *MC-Movie* contains 15,402 users, 3,048 critics, and 8,259 items with 261,292 user reviews and 144,541 critic reviews. *MC-Music* contains 11,483 users, 131 critics, and 5,133 items with 190,148 user reviews and 61,740 critic reviews. We can find that all the three datasets are extremely sparse in the user domain with the sparsity larger than 99.96%.

**Evaluation metrics.** To construct the training set, we randomly sampled 60% observed items for each user. Then, we sampled 10% observed items of each user for validation, and the rest data were used for the test. Hence, we randomly split each dataset five times and reported all the results by average values. We employed four widely used evaluation metrics for evaluating the performance, i.e., P@$K$, R@$K$, MAP@$K$, and NDCG@$K$ (Wu et al., 2018). For each user, P (Precision) @$K$ measures the ratio of correct prediction results among top-$K$ items to $K$ and R (Recall) @$K$ measures the ratio of correct prediction results among top-$K$ items to all positive items. Furthermore, MAP (Mean

---

[1]https://www.metacritic.com/
[2]Our collected datasets and all the code will be publicly available after the paper is accepted.

Table 1: The overall recommendation performances of different approaches.

| Datasets | Methods | R@5 | R@10 | P@5 | P@10 | MAP@5 | MAP@10 | NDCG@5 | NDCG@10 |
|---|---|---|---|---|---|---|---|---|---|
| *MC-Game* | CoNet | 0.0443 | 0.0733 | 0.0363 | 0.0308 | 0.0203 | 0.0126 | 0.0478 | 0.0578 |
| | DeepAPF | 0.0591 | 0.0934 | 0.0471 | 0.0381 | 0.0272 | 0.0166 | 0.0635 | 0.0747 |
| | NATR | 0.0804 | 0.1241 | 0.0635 | 0.0500 | 0.0385 | 0.0235 | 0.0873 | 0.1007 |
| | CMF | 0.0867 | 0.1364 | 0.0696 | 0.0560 | 0.0424 | 0.0263 | 0.0950 | 0.1107 |
| | DCDCSR | 0.0877 | 0.1376 | 0.0708 | 0.0567 | 0.0430 | 0.0269 | 0.0957 | 0.1118 |
| | SSCDR | 0.0913 | 0.1423 | 0.0742 | 0.0589 | 0.0452 | 0.0280 | 0.0994 | 0.1151 |
| | EMCDR | 0.0899 | 0.1408 | 0.0723 | 0.0572 | 0.0441 | 0.0272 | 0.0976 | 0.1136 |
| | BiTGCF | 0.0941 | 0.1474 | 0.0755 | 0.0601 | 0.0470 | 0.0292 | 0.1034 | 0.1203 |
| | MCIR | **0.1101** | **0.1650** | **0.0903** | **0.0695** | **0.0581** | **0.0358** | **0.1229** | **0.1388** |
| *MC-Movie* | CoNet | 0.0377 | 0.0652 | 0.0285 | 0.0250 | 0.0159 | 0.0103 | 0.0400 | 0.0488 |
| | DeepAPF | 0.0548 | 0.0925 | 0.0357 | 0.0308 | 0.0205 | 0.0132 | 0.0547 | 0.0665 |
| | NATR | 0.0706 | 0.1164 | 0.0434 | 0.0370 | 0.0256 | 0.0165 | 0.0689 | 0.0835 |
| | CMF | 0.0795 | 0.1285 | 0.0543 | 0.0456 | 0.0334 | 0.0218 | 0.0828 | 0.0977 |
| | DCDCSR | 0.0821 | 0.1331 | 0.0560 | 0.0467 | 0.0339 | 0.0222 | 0.0840 | 0.0993 |
| | SSCDR | 0.0884 | 0.1406 | 0.0614 | 0.0508 | 0.0378 | 0.0249 | 0.0898 | 0.1056 |
| | EMCDR | 0.0781 | 0.1258 | 0.0541 | 0.0450 | 0.0332 | 0.0216 | 0.0807 | 0.0954 |
| | BiTGCF | 0.0867 | 0.1394 | 0.0592 | 0.0488 | 0.0373 | 0.0240 | 0.0902 | 0.1059 |
| | MCIR | **0.1069** | **0.1586** | **0.0765** | **0.0594** | **0.0522** | **0.0331** | **0.1159** | **0.1299** |
| *MC-Music* | CoNet | 0.0420 | 0.0698 | 0.0280 | 0.0231 | 0.0155 | 0.0097 | 0.0396 | 0.0495 |
| | DeepAPF | 0.0526 | 0.0847 | 0.0337 | 0.0276 | 0.0186 | 0.0115 | 0.0492 | 0.0607 |
| | NATR | 0.0546 | 0.0896 | 0.0305 | 0.0255 | 0.0155 | 0.0096 | 0.0464 | 0.0590 |
| | CMF | 0.0784 | 0.1267 | 0.0505 | 0.0407 | 0.0315 | 0.0195 | 0.0773 | 0.0937 |
| | DCDCSR | 0.0878 | 0.1361 | 0.0545 | 0.0434 | 0.0353 | 0.0220 | 0.0844 | 0.1013 |
| | SSCDR | 0.0787 | 0.1238 | 0.0515 | 0.0407 | 0.0310 | 0.0193 | 0.0744 | 0.0897 |
| | EMCDR | 0.0910 | 0.1353 | 0.0569 | 0.0439 | 0.0362 | 0.0223 | 0.0866 | 0.1019 |
| | BiTGCF | 0.0894 | 0.1405 | 0.0574 | 0.0455 | 0.0365 | 0.0228 | 0.0867 | 0.1043 |
| | MCIR | **0.1176** | **0.1621** | **0.0735** | **0.0533** | **0.0547** | **0.0323** | **0.1199** | **0.1344** |

Average Precision) @$K$ and NDCG (Normalized Discounted Cumulative Gain) @$K$ consider the ranking of correct prediction results among top-$K$ items.

**Baselines.** We compare our proposed approach with various stat-of-the-art CDR methods. **CMF** (Singh & Gordon, 2008) is a classic collaborative CDR method. **CoNet** (Hu et al., 2018) and **DeepAPF** (Yan et al., 2019) utilize neural networks for CDR. **DCDCSR** (Zhu et al., 2020) considers the rating sparsity degrees of individual users in different domains. **SSCDR** (Kang et al., 2019) utilizes semi-supervised learning to map or share features. **EMCDR** (Man et al., 2017) combines Matrix Factorization and network-based bridging. **BiTGCF** (Liu et al., 2020) is a bi-directional transfer learning method that utilizes a Graph Collaborative Filtering network. **NATR** (Gao et al., 2021) is dedicatedly designed for the item-sharing scenario with neural transfer learning.

For the above-mentioned baselines, we utilized the open-source implementation provided by Recbole (Zhao et al., 2021) using PyTorch. Since most baselines were designed for user-sharing scenario, we extended them in a symmetrical manner to support item-sharing task. We used grid search for all the above baselines to carefully tune the corresponding parameters, such as the regularization coefficient and learning rate. In order to provide a fair comparison, we set the embedding size of all models to 150. Please see more implementation details in the Appendix.

## 4.2 OVERALL RECOMMENDATION PERFORMANCE

We present the overall recommendation performance results for the three datasets in Table 1 under two types of settings, i.e., $K = 5$ and $K = 10$. We can discover from Table 1 that MCIR can outperform all the baseline methods on every dataset owing to the multi-view learning framework and derived comprehensive representations. Specifically, MCIR outperforms the best baseline, by a relative boost of 19.60%, 17.00%, 23.61%, 18.86% for the metric P@5, R@5, MAP@5, and NDCG@5 in *MC-Game*, 20.93%, 24.59%, 39.09%, 29.06% in *MC-Movie*, and 29.23%, 28.04%, 49.86%, 38.29% in *MC-Music*, respectively. Hence, the results clearly demonstrate the effectiveness of our proposed approaches. Among the baseline methods, BiTGCF got the best performances in most conditions, maybe because its knowledge transfer module can effectively extend the flow of features from in-domain to inter-domain, and thus considers the integration of domain's common features and domain-specifc features. However, with only the graph view learning, BiTGCF cannot outperform our multi-view approach. Surprisingly, we find that CoNet, DeepAPF, and NATR performs not good in our experiments, even worse than CMF. A potential reason is that these three approaches were designed for the scenario that different domains share similar behavior patterns. While in our scenario, user and critics are quite different type in behaviors.

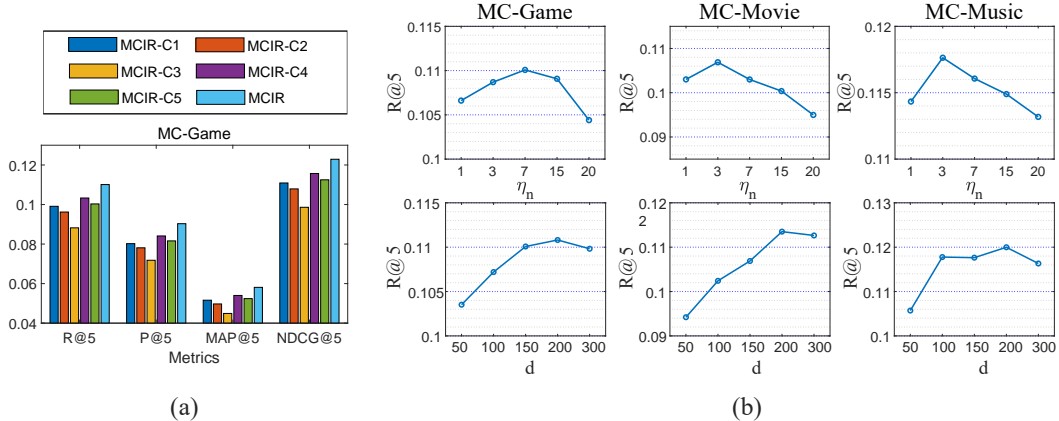

(a)                                                                                    (b)

Figure 3: Left: The ablation study of MCIR on the *MC-Game* Dataset. Right: The performance of R@5 with different values of hyper-parameter $\eta_n$ and dimension $d$ on the three datasets.

### 4.3 INVESTIGATIONS ON ABLATION STUDIES

In this section, we compare 5 variants of MCIR: MCIR-C1 without the review text information for user and critic embedding network; MCIR-C2 without the review text information for item embedding networks; MCIR-C3 without the entire critic domain; MCIR-C4 without the cross-view contrastive learning; MCIR-C5 with AE instead of VAE. Figure 3(a) presents the performance on *MC-Game* Dataset (More results on *MC-Movie* and *MC-Music* are in the Appendix). By comparing MCIR with MCIR-C1 and MCIR-C2, we can find that comment information is important for enhancing the performances. By comparing MCIR with MCIR-C3, we can find that the auxiliary information from critic domain is essential for recommendations in user domain. By comparing MCIR with MCIR-C4, we can validate the efficacy of our proposed cross-view contrastive learning. By comparing MCIR with MCIR-C5, we can observe that VAE can generate more accurate results than AE.

### 4.4 INVESTIGATIONS ON THE CROSS-VIEW CONTRASTIVE LEARNING MECHANISM

In this subsection, we assess the impact of the cross-view contrastive learning mechanism by adjusting the hyper-parameter $\eta_n$ within [1, 3, 7, 15, 20]. When $\eta_n = 0$, according to Equation 14, the contrastive learning mechanism is eliminated from the joint loss function. Figure 3 illustrates the performance of R@5 with varying values of the hyper-parameter $\eta_n$ across the three datasets. The performance results of other metrics can be found in the Appendix. The results depicted in Figure 3(b) reveal that the performance of MCIR initially improves as $\eta_n$ increases, validating that contrastive learning enables MCIR to learn the graph neighborhood information and thereby enhance its performance. However, when $\eta_n$ becomes excessively large, performance rapidly deteriorates. This is because the user-item-critic graph lacks rating information, so an overly large $\eta_n$ introduces noise rather than beneficial information.

### 4.5 INVESTIGATIONS ON THE DIMENSIONS OF THE LATENT SPACE

The number of dimensions $d$ is quite vital for the performance. If $d$ is too small, the latent space would have very weak representation ability to fit the real-world data. On the opposite, if $d$ is too large, the model complexity would also become too large and may face the over-fitting problem. The performance of R@5 with different values of dimension $d$ on the three datasets are presented in Figure 3(b) (see more in Appendix). We can observe that the performance result of MCIR is not good when $r = 50$. With a larger value of $d$, the performance tends to be much better. When $d = 200$, MCIR achieves the best results. With larger $d$, the performance of MCIR will begin to decrease.

### 5 CONCLUSION

In this work, we introduced the Multi-view Cross-domain Item-sharing Recommendation (MCIR) framework, an innovative approach to overcome the unique challenges posed in cross-domain recommendations with only item data being shared. MCIR adeptly amalgamates user preferences and critic opinions, employing distinct embedding networks explicitly designed for each perspective. Further, An attentive integration mechanism was designed to extract comprehensive item representations based on these multi-view representations. Moreover, we enhanced the framework by introducing a cross-view contrastive learning method, which harmonizes embeddings across different views by leveraging detailed neighborhood information from the user-item-critic graph. Finally, we conducted extensive experiments on three real-world datasets to validate the effectiveness of MCIR.

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

Table 2: The statistical information of the datasets.

| Dataset | MC-Game | MC-Movie | MC-Music |
|---|---|---|---|
| The number of items | 16,713 | 8,259 | 5,133 |
| The number of users | 18,622 | 15,402 | 11,483 |
| The number of critics | 522 | 3048 | 131 |
| The number of user-item interactions | 505,964 | 261,292 | 190,148 |
| The number of critic-item interactions | 242,764 | 144,541 | 61,740 |

## A  DATA DESCRIPTION

This appendix provides a more detailed data description of the datasets used in our experiments. The three datasets were collected from Metacritic[3], a website that aggregates reviews and assigns scores to various forms of media such as movies, TV shows, video games, music albums, and books.

Our data collection process spanned from the earliest records available on the website up until December 2022. We focused on three domains: movies, games, and music. Within each domain, we collected two types of data: user reviews and critic reviews.

For both user and critic reviews, we gathered the following information: 1. Ratings: We collected the rating assigned by each user or critic to an item. Ratings provide a numerical representation of their opinion or evaluation. 2. Review Text: We obtained the written reviews provided by users or critics for each item. These texts provide more detailed descriptions, opinions, or critiques. 3. Additionally, for each item, we collected the summary text, which briefly describes the item's information.

Table 2 presents some statistical information about the collected datasets. It is evident that all three datasets exhibit a high degree of sparsity. The user-item sparsity is measured at 99.84%, 99.79%, and 99.68% for MC-Game, MC-Movie, and MC-Music, respectively. Furthermore, the datasets vary in terms of the number of critics. The movie dataset has the highest number of critics, while the music dataset has the fewest. Comparatively, the number of critics is smaller than the number of users, but each critic tends to have a larger number of averaged interactions per critic than an individual user. Considering the data sparsity from the user perspective, leveraging critic information becomes crucial for addressing this issue and enhancing the comprehensiveness of recommendations.

We will make our collected raw data publicly available after the paper is accepted.

## B  EXPERIMENTAL SETTINGS

This section outlines the experimental settings employed in our study, including the preprocessing of text data, baseline methods, and implementation details.

### B.1  TEXT DATA PROCESSING

To ensure a fair comparison, all baseline methods utilized both rating data and text data. The text data were preprocessed by passing them through the pretrained representations provided by the model *down-paraphrase-multilingual-MiniLM-L12-v2* from HuggingFace Hub[4]. The obtained review representations were then incorporated into the baselines that did not initially contain a text embedding layer. We followed a consistent approach by utilizing the same embedding layer as our MCIR framework and integrating the review representations with the original baselines.

### B.2  BASELINE METHODS

We employed a set of baseline methods for comparison in our experiments. We utilized the open-source implementation provided by Recbole [5] using PyTorch, with certain methods extended symmetrically to accommodate the item sharing setting.

---

[3]https://www.metacritic.com/
[4]https://huggingface.co/models
[5]https://recbole.io/

We used grid search for all the above baselines to carefully tune the corresponding parameters. For CMF, we adjust the hyper-parameter $\alpha$ that controls the balance between source and target domain losses in the range $[0, 0.1, 0.2, ..., 0.8, 0.9, 1]$, and tune the regularization coefficients $\lambda$ and $\gamma$ for the source and target domains in the range $[0.001, 0.005, 0.01, 0.05, 0.1, 0.5, 1]$. For DCDCSR, we tune hyper-parameter $k$ for top-k similar items or users in the range $[1, 2, 5, 10]$. For SSCDR, we adjust the hyper-parameter $\lambda$ that controls the balance between supervised loss and unsupervised loss in the range $[0, 0.1, 0.2, ..., 0.8, 0.9, 1]$ and tune the the margin size in $[0.1, 0.2, 0.5, 1, 1.5]$. For BiTGCF, we tune the weight of the source or target embeddings in the transfer function in the range $[0.1, 0.2, ..., 0.8, 0.9]$ and set the number of GCF layers to 3 in order to achieve better performance. To ensure a fair comparison, we standardize the embedding size of all models to 150. Additionally, we perform tuning for the learning rate in the range of $[0.0001, 0.005, 0.001, 0.005, 0.001]$, and the L2 normalization coefficient in the range of $[1e^{-1}, 1e^{-2}, 1e^{-3}, 1e^{-4}, 1e^{-5}]$. For CoNet EMCDR, DCDCSR, and SSCDR, we employ the same fully connected layer consisting of two layers with a dimensionality of 150.

### B.3 IMPLEMENTATION DETAILS

We have included the code as part of the supplementary materials accompanying our paper. The code will be made publicly available after the acceptance of the paper.

To ensure a fair comparison, we set the number of dimensions $d$ to 150 for the MCIR framework. This dimensionality was chosen to align with the standard configuration across the baseline methods. We set the dropout ratio as 0.5 for the user-rating embedding network and 0.2 for the attentive integrated mechanism. We tune the regularization hyper-parameter $\lambda$ in $[1, 5, 10, 15, 20]$. After experimentation, we found that $\lambda = 15$ yielded the best performance. We tuned the hyper-parameters of each loss in [1, 5, 10, 15, 20]. Specifically, we found the following values yielded the best performance: $\eta_s$ best value: 10, $\eta_c$ best value: 10, and $\eta_n$ best value: 3. These hyper-parameters control the relative importance of each loss component during training. For the user-rating view, we adopted a 2-layer MLP network as the encoder layer of VAE and used *tanh* as the activation function. Moreover, to prevent the posterior collapse of the VAE, we adopted the warm-up trick. This involves gradually increasing the weight of the KL-divergence term in the VAE loss from 0 to 1 during training. The warm-up schedule was implemented batch-wise. For the critic view, we set the number of maximized sampled critics to 20. This value determines the number of critics that are considered each batch. For the graph view, we set the number of GCN layers to 3 for effective aggregation and propagation of information across the graph structure. In the contrastive learning component, we set the temperature hyper-parameter $\tau$ to 0.2. This parameter controls the smoothness of the probability distribution used in the contrastive loss.

## C EXPERIMENTAL RESULTS

In this section, we first supplement the figures for the results in Section 4.3, 4.4 and 4.5 in the paper. Then we show some case studies.

### C.1 INVESTIGATIONS ON ABLATION STUDIES

In this subsection, we conducted investigations on the ablation studies, which can demonstrates the necessaries of different components in our proposed MCIR framework. As discussed in the paper, we compare MCIR with five variants on the three datasets. Figure 4 presents the performance of different variants of MCIR on the *MC-Movie* and *MC-music* datasets. In comparing MCIR with MCIR-C1 and MCIR-C2, it becomes evident that incorporating comment information significantly bolsters performance. A comparison between MCIR and MCIR-C3 underscores the indispensable role of auxiliary information from the critic domain in enhancing recommendations in the user domain. The juxtaposition of MCIR with MCIR-C4 unequivocally validates the effectiveness of the proposed cross-view contrastive learning approach. Lastly, by contrasting MCIR with MCIR-C5, we ascertain that VAE consistently outperforms AE, delivering more precise results.

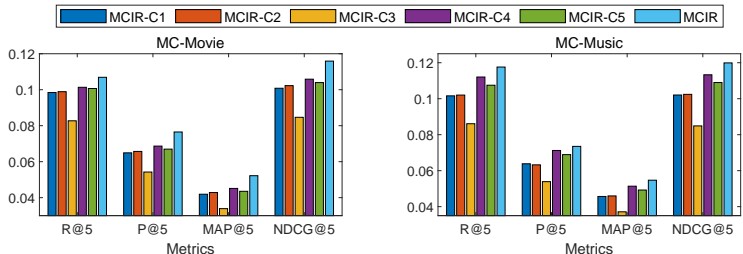

Figure 4: The performance of different variants of MCIR on the *MC-Movie* and *MC-music* datasets.

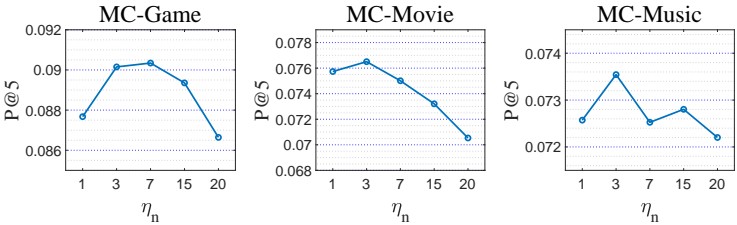

Figure 5: The performance of P@5 with different values of hyper-parameter $\eta_n$ on the three datasets.

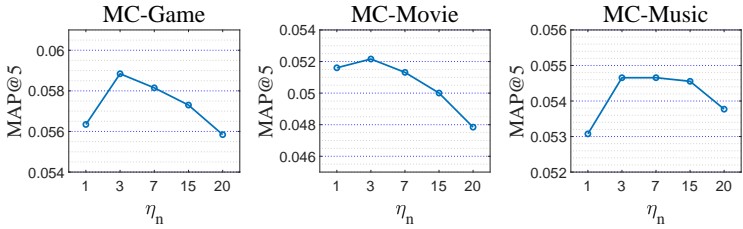

Figure 6: The performance of MAP@5 with different values of hyper-parameter $\eta_n$ on the three datasets.

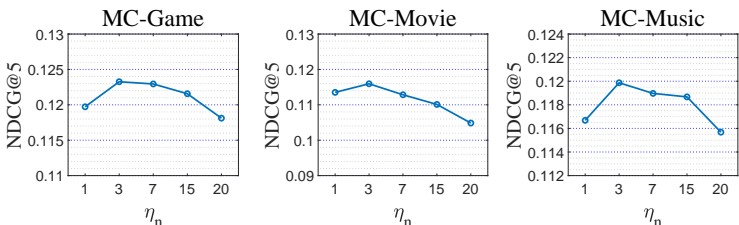

Figure 7: The performance of NDCG@5 with different values of hyper-parameter $\eta_n$ on the three datasets.

### C.2 INVESTIGATIONS ON THE CROSS-VIEW CONTRASTIVE LEARNING MECHANISM

In this subsection, we conducted investigations on the hyper-parameter $\eta_n$, which controls the cross-view contrastive learning mechanism in the MCIR framework. We present the performance of P@5, MAP@5, and NDCG@5 with varying values of the hyper-parameter $\eta_n$ across the three datasets in Figure 5, 6, and 7. We observed that the best performances were achieved when $\eta_n = 3$. These findings support the conclusion presented in the paper, highlighting the importance of the cross-view contrastive learning mechanism in leveraging neighborhood view information. It is worth noting that without the cross-view contrastive learning mechanism, the MCIR framework may not effectively utilize the neighborhood view information. However, setting $\eta_n$ to excessively large values may introduce more distractive information, leading to decreased performance. Therefore, careful

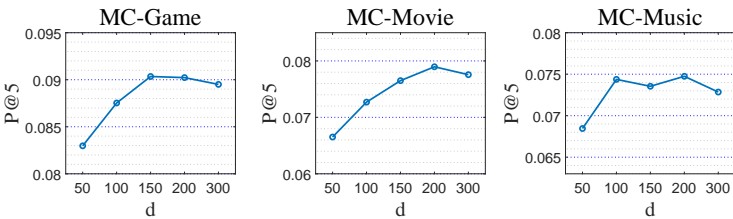

Figure 8: The performance of P@5 with different values of dimension $d$ on the three datasets.

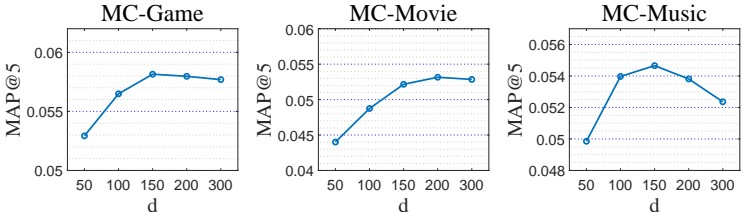

Figure 9: The performance of MAP@5 with different values of dimension $d$ on the three datasets.

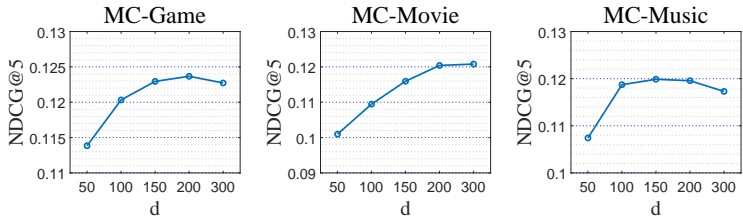

Figure 10: The performance of NDCG@5 with different values of dimension $d$ on the three datasets.

selection of $\eta_n$ is crucial to strike a balance between exploiting useful information and avoiding distraction from the graph view.

### C.3 INVESTIGATIONS ON THE DIMENSIONS OF THE LATENT SPACE

n this subsection, we investigated the impact of the dimensionality $d$ of the latent space on the performance of the MCIR framework. We evaluated the performance of P@5, MAP@5, and NDCG@5 across the three datasets while varying the value of $d$. Figure 8, 9, and 10 depict the results obtained with different values of $d$. Our experiments revealed that the performance of MCIR initially increased and then decreased with the growth of $d$. This behavior can be attributed to the following reasons: If $d$ is not large enough, the model may struggle to learn effective representations that capture the underlying patterns and nuances in the data, resulting in suboptimal performance. However, if $d$ is excessively large, the model may face the risk of overfitting, where it becomes overly specialized to the training data and fails to generalize well to unseen data. This can lead to a decrease in performance on the evaluation metrics. Hence, there is a trade-off in selecting the appropriate dimensionality for the latent space in the MCIR framework. The optimal value of $d$ will strike a balance between capturing sufficient information and avoiding over-fitting.

## D CASE STUDY

In this section, we provide case studies to demonstrate the explainability of our attentive integration mechanism. This mechanism is designed to extract comprehensive item representations by considering the multi-view representations and calculating attention scores between the item summary and critic review text.

Table 3: Analysis of attentive integration mechanism: case study with attention scores for item summary and critic reviews

| Item | CALM (by 5 Seconds of Summer) | |
|---|---|---|
| The average user score on Metacritic | 9.0 out of 10 | |
| The average critic score on Metacritic | 70.00 out of 100 | |

| Critics | Rating value | Attention score |
|---|---|---|
| summary | - | 0.0193 |
| The Line of Best Fit | 80 out of 100 | 0.2501 |
| AllMusic | 80 out of 100 | 0.2184 |
| No Ripcord | 80 out of 100 | 0.1940 |
| Clash Music | 60 out of 100 | 0.1848 |
| Rolling Stone | 50 out of 100 | 0.1333 |

| Item | NCAA FOOTBALL 12 (PlayStation 3) | |
|---|---|---|
| The average user score on Metacritic | 5.8 out of 10 | |
| The average critic score on Metacritic | 82.0 out of 100 | |

| Critics | Rating value | Attention score |
|---|---|---|
| summary | - | 0.7446 |
| PlayStation Universe | 85 out of 100 | 0.0469 |
| Playstation Official Magazine UK | 70 out of 100 | 0.0464 |
| Digital Chumps | 90 out of 100 | 0.0436 |
| PlayStation LifeStyle | 80 out of 100 | 0.0424 |
| TotalPlayStation | 90 out of 100 | 0.0382 |
| Playstation: The Official Magazine (US) | 90 out of 100 | 0.0379 |

Table 3 presents two typical cases from our datasets, including the corresponding user/critic ratings and attention scores.

The first case is about the music album *CALM* by the Australian pop band *5 Seconds of Summer*. The critics give some comments from the professional perspective and finally the average score is not large. However, the user reviews, which mostly come from fans, gave significantly larger average scores. Many users believe the albumn "is so unique and so unbelievably good". In this scenario, it is important for the recommendation model to understand the divergence between user and critic perspectives and extract the most relevant information from critic reviews to enhance recommendations for users. From Table 3, we can observe that MCIR assigns higher attention scores to critics with high ratings and lower attention scores to critics with low ratings. This allows MCIR to extract more useful information from the critic reviews through user-experience-related perspective. Additionally, MCIR assigns a very small attention score to the summary text, as the summary of the album is simple and contains little information.

The second case revolves around the video game "NCAA FOOTBALL 12 (PlayStation 3)". The critics gave this item a high rating, with an average score of 84.17 out of 100, stating that it is "a must-have for college football fans". However, the user reviews present a different perspective, with an average score of 6.5 out of 10. Users also find the game is "a fun and addicting to the NCAA franchise" but criticize its lack of innovation compared to *NCAA FOOTBALL 11*. Our MCIR approach effectively captures the difference between the user and critic perspectives through multi-view learning. Consequently, our approach assigns lower attention scores to the critics' reviews and a higher score to the summary text.

