# OpenReview forum: "Unifying User Preferences and Critic Opinions: A Multi-View Cross-Domain Item-sharing Recommender System"
_ICLR.cc/2024/Conference — Submitted to ICLR 2024_

### Official Review · Reviewer_c3D1 · 2023-10-24

**Soundness:** 2 fair
**Presentation:** 2 fair
**Contribution:** 2 fair
**Rating:** 3
**Confidence:** 4

**Summary:**

In this paper, the authors propose a Multi-View Cross-domain Item-sharing Recommendation (MCIR) framework that synergizes user preferences with critic opinions. The proposed MCIR achieves state-of-the-art performance on several real-world datasets.

**Strengths:**

- Utilizing critic review information for knowledge sharing across domains is interesting.

- The proposed MCIR achieves state-of-the-art performance on several real-world datasets.

**Weaknesses:**

- There are many incorrect formula details in the article. For example, the authors obtain covariance $\Sigma$ via $W_T’h_T + b_T’ = diag(\Sigma_t)$. However, $\Sigma$ should be positive values. How to guarantee the output of $W_T’h_T + b_T’$ could be always positive?

- Some model details are missing. For example, the authors adopt $g(\cdot)$ as the activation function in Eq.(1). However, what kind of activation function do you use in the experiments? ReLU or Sigmoid? Moreover, the authors adopt the dropout layer in the Eq.(1). How about the dropout ratio for this layer?

- Some formulas are completely wrong. For example, the authors obtain user distribution as $p(u_i|R_i) \sim N(\mu_i, \Sigma_i)$. However, the reparameterization process in the paper is $u_i = \mu_i+\epsilon \Sigma_i$. It is completely wrong. The correct answer is $u_i = \mu_i+\epsilon \sqrt{\Sigma_i}$.

- The methodology is hard to follow with too many notations. I strongly encourage the authors to provide the pseudo algorithm table.

- Some important baselines are missing, e.g., PTUPCDR.

- The authors emphasize that critic reviews are much more valuable than common reviews. However, how to define whether the reviews are critic or not?

- The dataset statistics are missing key information, e.g., number of overlapped users.

- Can the proposed method handle different ratios of overlapped users (e.g., only 10% users are overlapped across domains)?

======
Update:

I acknowledge that I have read the authors response. Although the idea of this paper is interesting, it still needs major revision on technically details (e.g., formula and symbol corrections) to make it more precise.

**Questions:**

Please refer to the Weakness above.

---

> ### Author Response · Authors · 2023-11-15
> **Response to Reviewer c3D1**
>
> We are thankful for your review and the opportunity to clarify aspects of our paper. Here are our responses to your concerns:
>
> 1. Sorry for the confusion on data. First we want to clarify that, as detailed in Sections 1 and 4.1, and the Appendix, critic and user reviews in our dataset, sourced from Metacritic, are distinctly labeled, eliminating the need to differentiate between expert and amateur reviews. This dataset will be made publicly available upon paper acceptance.
>
> 2. Also we want to clarify that there are no user overlaps in our datasets. As stated in the paper, our work focuses on cross-domain recommendations where different domains share items but not users or critics. The cross-domain in this paper means we leveraging auxiliary information from critic domain data for making better predictions in the user domain. Here 'domain' is defined on the item-level relevance, i.e., there are common items and different users between the targeted and auxiliary domains, as illustrated in the survey paper [1,2]. The auxiliary domain is critic domain and the targeted domain is user domain in this paper. We concur that this specific area remains relatively untapped compared to traditional user-sharing cross-domain recommendation, thus underscoring the significance of our contribution. We will add more explanation in the paper for revision.
>
> The following is our responses to your questions:
>
> Weakness 1. The confusion regarding $diag{\Sigma_{i}}$ is regrettable. In our code, $W_{T}'h_{T}+b_{T}'$ outputs the log variance, which we then convert to positive variance values using the 'torch.exp' function. This is a very frequently used trick in VAE style code. Since this is a common code trick, we did not mention it in the paper. We will add more explanations in the Appendix for revision.
>
> Weakness 2. The activation function in Eq.(1) is tanh in our experiments. We will add more details about it for revision.
> As introduced in Section 4.1, we set the dropout ratio as 0.5 for the user-rating embedding network.
>
> Weakness 3. We appreciate your correction on the reparameterization formula. This will be amended in the revised paper.
>
> Weakness 4. Thanks for your suggestion. We will incorporate a pseudo algorithm table in the revised paper.
>
> Weakness 5. PTUPCDR assumes that there is overlapping users and no shared items, which is opposite to our focused scenario, i.e., item-sharing and no user overlaps. Therefore, PTUPCDR cannot be directly applied in our problem settings and not included in our baseline methods.
>
> Weakness 6. As mentioned, the distinction between critic and user reviews is inherent in our dataset with labels, negating the need for manual detection.
>
> Weakness 7. In our experiments, there are no overlapping users.
>
> We sincerely appreciate your time and effort in reviewing our work. We hope our responses have satisfactorily addressed your queries. Given the initial misunderstanding regarding the data and cross-domain definition, if you find our clarifications adequate, we kindly request you to consider revising the rating for our paper.
>
> [1] Zang T, Zhu Y, Liu H, et al. A survey on cross-domain recommendation: taxonomies, methods, and future directions[J]. ACM Transactions on Information Systems, 2022, 41(2): 1-39.
>
> [2] Zhu F, Wang Y, Chen C, et al. Cross-domain recommendation: challenges, progress, and prospects[J]. arXiv preprint arXiv:2103.01696, 2021.

---

> > ### Author Response · Authors · 2023-11-21
> >
> > Dear Reviewer c3D1,
> >
> > I hope this email finds you well. Firstly, I'd like to express our gratitude for the time and effort you've put into reviewing our paper.
> >
> > We have thoroughly addressed the comments and concerns raised in your review. Given that there may have been some misunderstandings, it's crucial for us to understand your perspective after reading our explanations.
> >
> > Thank you in advance for your time and cooperation.
> >
> > Best regards

---

### Official Review · Reviewer_3TsF · 2023-11-01

**Soundness:** 3 good
**Presentation:** 2 fair
**Contribution:** 3 good
**Rating:** 5
**Confidence:** 4

**Summary:**

The authors introduced a multi-view cross-domain item-sharing recommendation algorithm to involve critic comment from users. They involved many techniques (e.g., GCN and contrastive learning etc) to obtain the user item aligned embeddings. Various baseline methods and datasets were investigated in the experiments. Results show the advantages of the proposed.

**Strengths:**

S1. critic comment is a good idea to enhance recommendation in general

S2. sufficient baselines were selected in experiments

S3. Comprehensive experiments were conducted

**Weaknesses:**

W1. hard to follow the methodology

W2. unclear definitions

W3. critic comment is the one of the main contribution but it's not clear about how to define and how to detect critic comments

**Questions:**

Q1. It's hard to follow the methodology sections. Figure 2 doesn't help make it clearer. Instead it makes it more complicated to understand without knowing the meaning of letters and captions. I hope reading through text would help my understanding. However I still don't know why many steps are necessary and why so much components and techniques are required. For example, for definitions, what user-rating network, user-comment network, and critic embedding network. It seems that the final goal is to obtain user and item embeddings aligned in the same latent space. But by nature critic comment is hard to define and detect (please refer to the following question). It seems that contrastive loss and GCN are also included, but by checking Figure 2 are the left and right boxes decoupled? Do they have relationship? And what's their relationship?

Q2. “Experts Write More Complex and Detached Reviews, While Amateurs More Accessible and Emotional Ones” How to know which review is written by an expert or an amateur? Or how to quantify the critic metric for a comment. The statement is also related to Section "Critic Embedding Network" where it mentioned critic rating prediction task. It seems to be based on the inner product of v_j and w_l^c. What does the latent critic-rating vector w_j^c come from?

==========================
I acknowledge that I have read the authors response. I appreciate the authors efforts. But it didn't address my concerns. I would keep my original rating.

---

> ### Author Response · Authors · 2023-11-15
> **Response to Reviewer 3TsF**
>
> Thank you for your response. We have checked them carefully and we have the following responses:
>
> 1. Sorry for the confusion on data. First we want to clarify that, as detailed in Sections 1 and 4.1, and the Appendix, critic and user reviews in our dataset, sourced from Metacritic, are distinctly labeled, eliminating the need to differentiate between expert and amateur reviews. This dataset will be made publicly available upon paper acceptance.
>
> 2. We apologize for any prior ambiguities regarding the components of the MCIR framework.
> The MCIR framework is composed of four integral components, each contributing significantly to the methodology:
>
> * User Embedding Networks: This includes two sub-components - the user-rating embedding network and the user-comment embedding network. Their primary function is to learn latent user embeddings from both rating and comment perspectives, respectively.
>
> * Critic Embedding Networks: These networks are designed to concurrently learn latent critic embeddings from both rating and comment perspectives. This dual approach ensures a comprehensive representation of critic opinions.
>
> * Attentive Integrated Mechanism: This mechanism focuses on learning latent item embeddings. It plays a crucial role in transferring auxiliary critic text information into the user domain by enabling shared item embeddings.
>
> * Cross-View Unified Learning with Contrastive Learning: This component is essential for leveraging neighborhood information across user and critic domains. The incorporation of contrastive learning here enhances the effectiveness of using cross-domain neighborhood information to refine the latent user and item embeddings.
>
> The attentive integrated mechanism and cross-view unified learning collectively ensure a robust transfer and utilization of critic information from both the text and graph views.
> We have provided a comprehensive ablation study in Section 4.3 and the Appendix, which demonstrates the importance of each component.
> In response to your concerns, we will clarify the main components of the MCIR framework and add more intuitive and descriptive explanations when introducing the modules' details in the revised version of our paper.
>
> 3. Regarding Figure 2, the left and right boxes are interconnected via the final latent embeddings $p_i$ and $q_j$, which are trained through both the VAE loss in Equation 9  and the contrastive learning loss in Equation 13.
> We will try to modify the Figure 2 for better illustration.
>
> 4. As outlined in Section 3.3, the latent critic-rating vector $w_l^c$ is derived from the latent critic vector $u_l^c$ using a single-layer MLP network ($w_l^c = f_w(u_l^c)$). The vector $u_l^c$ is initially randomly initialized and then refined through iterative learning.
>
> It appears that there may be some misunderstandings about our work. We sincerely hope that our responses have addressed your concerns and clarified any points of confusion. If you have any further questions or require additional clarification, please do not hesitate to reach out to us.

---

> > ### Author Response · Authors · 2023-11-21
> >
> > Dear Reviewer 3TsF,
> >
> > I hope this email finds you well. Firstly, I'd like to express our gratitude for the time and effort you've put into reviewing our paper.
> >
> > We have thoroughly addressed the comments and concerns raised in your review. Given that there may have been some misunderstandings, it's crucial for us to understand your perspective after reading our explanations.
> >
> > Thank you in advance for your time and cooperation.
> >
> > Best regards

---

### Official Review · Reviewer_Pmes · 2023-11-07

**Soundness:** 3 good
**Presentation:** 2 fair
**Contribution:** 3 good
**Rating:** 5
**Confidence:** 3

**Summary:**

The authors proposed to explore a less explored scenario: cross-domain recommendation with distinct user groups, sharing only item-specific data. Towards this end, they proposed a multi-view cross-domain item-sharing recommendation framework that leverages user reviews, critic comments, item summaries, user ratings, and critic ratings. They collected a dataset with three domains, namely Game, Movie, and Music from Metacritic and compared with multiple baselines.

**Strengths:**

1. The cross-domain recommendation problem with distinct users across domains is an interesting yet overlooked problem. The authors took a look at this problem and proposed a complicated model that uses multiple types of data to solve it.
2. The proposed model showed good performance under the authors' setting and outperformed all baselines. Modeling-wise, this provides some insight into what is worth trying and effective in similar problems and can inspire more innovative solutions.

**Weaknesses:**

1. First of all, the authors did not do a good job of clearly formulating the problem they want to solve. I felt confused after I went over the paper for the first two times. When I first read it, I thought they were trying to do cross-domain recommendations when different domains shared items but not users, which is counter-intuitive. Then I realized it's critic that is shared by different domains. I recommend the authors state this very explicitly and use some space to formulate the problem using some formulas. Besides, the phrase "item-sharing" in the title is quite misleading.
2. In experiments, did the authors use data from two datasets for training and then predict the rating for the 3rd dataset? How does the evaluation of "Cross-domain" recommendation work in this paper? I did not quite understand after reading Section 4.1.
3. Besides the experiment section, this paper also needs more clarity in its description of the proposed model can be further improved to get better readability. Questions related to this can be found in the next section.

**Questions:**

1. Shouldn't the $y_{lj}$ in the left up corner of Figure 2 be $y^c_{lj}$?
2. In the user embedding network, why is $R_i$ used to construct $u_i$ instead of the other way around, using $u_i$ to generate $R_i$? The authors may have their rationale for designing the model in this way. But they failed to explain it clearly to the readers.
3. The explanation of the "attentive integrated mechanism" seems to be over-complicated to me. It actually follows the standard design of the attention mechanism with $v_j$ as the query vector and $w^c_l$ as the key and value vector. The introduction of $L_j$ and $w^c_0 = v_j$ does not seem to be necessary and only added complexity.
4. In the same section, the first sentence says "Given that only the items are shared between the critic and user domains". I believe the word "domain" does not mean the cross-domain studied in this paper is trying to cross the "user" and "critic" domains.  Am I right?

---

> ### Author Response · Authors · 2023-11-15
> **Response to Reviewer Pmes**
>
> We are grateful for the opportunity to clarify key points in our paper. Below, we address your concerns and provide further explanations:
>
> 1. First, we are sorry for any ambiguity regarding the term 'cross-domain' in our context. Our work indeed focuses on cross-domain recommendations where different domains share items but not users or critics. Critics are not shared by different domains.
>
> Traditionally, cross-domain recommendations have been usually defined in the way that users are sharing across different domains in the literature. However, as evidenced by the survey paper [1] and [2] (please see the Figure 1 and Table 1 in [1]), it's essential to recognize that recommendations integrating distinct user groups with full item overlap also fall under the cross-domain recommendation umbrella [1][2][3]. As stated in the paper, the cross-domain in this paper means we leveraging auxiliary information from critic domain data for making better predictions in the user domain. Here 'domain' is defined on the item-level relevance, i.e., there
> are common items and different users between the targeted and auxiliary domains, as illustrated in the survey paper [2]. The auxiliary domain is critic domain and the targeted domain is user domain in this paper. We concur that this specific area remains relatively untapped compared to traditional user-sharing cross-domain recommendation, thus underscoring the significance of our contribution. We will add more explanation in the paper for revision.
>
> 2. In our experiments, we train and test the model on each dataset independently. There is no data sharing among the three datasets. MCIR can outperform all the baseline methods, demonstrating the effectiveness of our cross domain leaning approach.
>
> The following is our responses to your questions:
>
> Question 1. Thank you. $ y_{lj} $ should be $ y_{lj}^c $ in the left up corner of Figure 2. We will correct this in the revised paper.
>
> Question 2. As stated in the paper, the user embedding network belongs to the encoder part of the VAE architecture. Hence we use $R_i $ to construct $ u_i $ but not using $ u_i $ to generate $R_i $. The latter is suitable for the decoder part of VAE. We will add more explanation in the revised paper.
>
> Question 3. Sorry for the confusion around the introduction of $L_j$ and $w_{0}^c=v_j$ in Equation 6. These elements are meant to simplify the mathematical expression. Without the help of $ L_j $ and $ w_{0}^c=v_j $, Equation 6 will look much more complex. According to your comment,  we will refine our explanation of the attentive integration mechanism, particularly focusing on the roles of key, query, and value vectors.
>
> Question 4.  As previously mentioned, our study defines 'cross-domain' as leveraging critic domain data to enhance user domain predictions. Critics, often from professional media institutions, and users, typically ordinary web contributors, represent distinct groups without overlap. Our study bridges these 'user' and 'critic' domains in a novel cross-domain approach.
>
> We sincerely appreciate your time and effort in reviewing our work. We hope our responses have satisfactorily addressed your queries. Given the initial misunderstanding regarding the cross-domain definition, if you find our clarifications adequate, we kindly request you to consider revising the rating for our paper.
>
> [1] Zang T, Zhu Y, Liu H, et al. A survey on cross-domain recommendation: taxonomies, methods, and future directions[J]. ACM Transactions on Information Systems, 2022, 41(2): 1-39.
>
> [2] Zhu F, Wang Y, Chen C, et al. Cross-domain recommendation: challenges, progress, and prospects[J]. arXiv preprint arXiv:2103.01696, 2021.
>
> [3] Gao C, Li Y, Feng F, et al. Cross-domain recommendation with bridge-item embeddings[J]. ACM Transactions on Knowledge Discovery from Data (TKDD), 2021, 16(1): 1-23.

---

> > ### Comment · Reviewer_Pmes · 2023-11-19
> >
> > Thank you for the responses.
> >
> > to Answer 1: can you list a few concrete examples of "cross-domain recommendations where different domains share items but not users"? Asking because examples can help readers build an intuitive understanding of why this is an important problem to study and why it is challenging.
> >
> > to Answer 2:  did you mention you "train and test the model on each dataset independently"? If yes, can you point me to it? If not, it's necessary to add it.
> >
> > Thanks!

---

> > > ### Author Response · Authors · 2023-11-20
> > >
> > > Thank you for your questions, which provide an opportunity to further elucidate our work.
> > >
> > > Response to Question 1:
> > > Cross-domain recommendations where different domains share items but not users are particularly relevant in situations where user groups have distinct characteristics, or when there's a need for privacy protection and platform-specific data policies. For instance:
> > >
> > > * In our study, we examine the scenario of user and critic reviews. Research in management and marketing [1,2,3] highlights significant differences between ordinary web user reviewers and professional critics. Mixing these two groups for a single model training is not ideal due to these inherent differences. Critics generally provide insights more closely related to the item properties, while users often share personal experiences [1,2]. By leveraging information from the critic domain, we can enhance item recommendations in the user domain, especially in scenarios where the user domain lacks data but the critic domain is resource-rich, such as pre-release product reviews by media organizations. To solve the problem, our approach involves distinct embedding networks for users and critics, bridged through an attentive mechanism and contrastive learning to effectively transfer knowledge across domains.
> > >
> > > * Gau et al. (2021)[4] and Gau et al. (2019)[5] explored scenarios involving different companies or organizers, where sharing user behavior data is constrained by company policies, yet item information can be shared. This situation is common in overlapping services like hotels on Trip.com and Booking.com, or movies on IMDb and Douban. Their work on datasets like ML-NF (MovieLens and Netflix) and TC-IQI (iQiyi and Tencent Video) demonstrates the practical importance of item-sharing cross-domain recommendations.
> > >
> > > These examples illustrate the real-world relevance and challenges of item-sharing cross-domain recommendations, underscoring the importance of our research focus.
> > >
> > > Response to Question 2:
> > > Yes, we independently train and test the model on each dataset. For every dataset, critic-item reviews (ratings and texts) serve as auxiliary domain data, while user-item reviews (ratings and texts) form the targeted domain data. As introduced in Section 4.1, we used 60\% of observed items per user for training, 10\% for validation, and the remaining for testing in the targeted domain, while all auxiliary domain data were used for training. We evaluate our model on the test set in the targeted domain. Many thanks for your question, and we will include more detailed descriptions of our experimental setup in the revised paper.
> > >
> > > [1] Santos T, Lemmerich F, Strohmaier M, et al. What's in a review: Discrepancies between expert and amateur reviews of video games on metacritic[J]. Proceedings of the ACM on human-computer interaction, 2019, 3(CSCW): 1-22.
> > >
> > > [2] Dillio R. Different Scores: Video Gamers' Use of Amateur and Professional Reviews[M]. Rochester Institute of Technology, 2013.
> > >
> > > [3] Parikh A A, Behnke C, Almanza B, et al. Comparative content analysis of professional, semi-professional, and user-generated restaurant reviews[J]. Journal of foodservice business research, 2017, 20(5): 497-511.
> > >
> > > [4] Gao C, Li Y, Feng F, et al. Cross-domain recommendation with bridge-item embeddings[J]. ACM Transactions on Knowledge Discovery from Data (TKDD), 2021, 16(1): 1-23.
> > >
> > > [5] Gao C, Huang C, Yu Y, et al. Privacy-preserving cross-domain location recommendation. Proceedings of the ACM on Interactive, Mobile, Wearable and Ubiquitous Technologies, 2019, 3(1): 1-21.

---

> > > > ### Comment · Reviewer_Pmes · 2023-11-20
> > > >
> > > > Thanks for the detailed response! I'd like to move up my rating. The authors did propose a very effective approach, however, the paper overall, especially its presentation, may not be good enough for ICLR yet.

---

> > > > > ### Author Response · Authors · 2023-11-20
> > > > >
> > > > > Thank you very much for acknowledging 'The authors did propose a very effective approach' and for your decision to adjust the rating upward. We are heartened by your recognition of the value of our work.
> > > > >
> > > > > We understand your concerns regarding the presentation of our paper, particularly in how our problem setting diverges from typical cross-domain scenarios. To remedy this, we are fully committed to enhancing the clarity and overall presentation in our revised paper.
> > > > >
> > > > > Should you have any further questions or require additional details on any aspect of our study, please do not hesitate to contact us. We are eager to engage in further discussions and clarify any aspects of our work, with the goal of strengthening our paper and potentially improving its rating.
> > > > >
> > > > > Once again, thank you for your feedback.

---

### Official Review · Reviewer_6fW2 · 2023-11-08

**Soundness:** 3 good
**Presentation:** 4 excellent
**Contribution:** 2 fair
**Rating:** 5
**Confidence:** 3

**Summary:**

The paper builds a cross domain recommender system that leverages ratings and reviews from users belonging to different groups, and item descriptions. In particular, the different groups share no common users while makes the problem of information transfer across them more difficult. The paper builds an elaborate system with multiple components to overcome this challenge and obtains SOTA results.

**Strengths:**

1. The problem is well motivated
2. The paper is written clearly
3. The results beat SOTA by a good margin on the given datasets

**Weaknesses:**

The centra weakness is the lack of a detailed study of its individual components. For a system, that has as many components as this one, it is crucial to measure the relative importance of each. While the paper has some study ablation studies I do not think they are exhaustive (please see questions below). For instance,

1. How useful is the attentive module? For instance what if all $v^a_j = 0$?
2. How useful is the graph network $\left(\eta_3 = \eta_4 = 0\right)$  ?
3. Why not also use a VAE in the critic embedding network for training $u_l^c$. Conversely why not use the (analogous) first term in equation of 5 in equation 3, and just not use the VAE at all? Could you please elaborate.


It is difficult to assess the impact of the proposed method (in terms of scope, generalizability, etc.) without understanding the value of its components.

**Questions:**

1. What are $\eta_n,\eta_s, \eta_c$? From the context I seem to gather that $\eta_n = \eta_4$?

2. What does ablation study C1 mean? Does that mean neither critic comments or user comments or item summary text are used?

3. Similarly, what does C2 mean?

4. Does C3 mean making $\mathcal L_{Multi} = 0, v^a_j = 0$ and excising all the critic-item edges from the graph?

5. Does C4 mean, $\eta_4 =0 $?

6. How does the proposed method compare with the best prior methods (say BitGCF, EMCDR, SSCDR) in terms of training time, number of parameters, and inference cost?

7. [Minor] In equation 7, one could catenate and then project? Do you think that could lead to substantial gains?

8. [Minor] Is there a reason the decoder network is chosen to be as simple as eq. 8? Needless to say, simple is good. But wondering if there are other motivations.

9. [Broad] It seems there are existing datasets used by prior baselines (ML-NF dataset). Is there a reason for not choosing it over Metacritic (or for that matter, using both)?

For all the ablation questions, please answer in terms of what happens to equation 14, wherever possible.

---

> ### Author Response · Authors · 2023-11-15
> **Response to Reviewer 6fW2**
>
> We greatly appreciate your insightful feedback and recognition of the strengths in our paper, particularly the well-motivated problem statement and our system's ability to surpass state-of-the-art results. Below, we have addressed your concerns in detail:
>
> Weakness 1: The attentive integration mechanism is essential in combining item-summary information with critics' commentary as it adaptively aggregates information, reflecting the varying influences of item summaries and critics’ comments on item characteristics. We have included two case studies in Appendix Section D, demonstrating its effectiveness. In response to your suggestion, we introduce a new variant, MCIR-C6, employing simple average of item-summary vector $v_j^s$ and critic vectors $w_{l}^c$ instead of attentive integration. The comparative results on MC-Game are as follows:
>
> Methods R@5 P@5 MAP@5 NDCG@5
>
> MCIR-C6 0.1050 0.0845 0.0545 0.1172
>
> These results clearly show the superior performance of MCIR over MCIR-C6, affirming the value of the attentive integration mechanism. Regarding the query “what if all $v_j^a=0$?”, in practice, $v_j^a$ will not be zero since the item-summary vector is always present. The variant MCIR-C2, which replaces $v_j^a$ with the item-summary vector $v_j$, underperforms compared to MCIR as shown in Figure 3(a). This further highlights the importance of the review text information alongside the attentive integration mechanism. We plan to include MCIR-C6 in our revised ablation study and provide an in-depth discussion of the attentive integration mechanism.
>
> Weakness 2: Actually, the variant MCIR-C4 is set as $\eta_3=\eta_4=0$. Note that the graph embeddings can only influence the final latent user and item representation $p_i $ and $q_j $ by the contrastive learning loss  $ \mathcal{L}_{CL} $.
>
> So when we set $\eta_4=0$, the GCN loss $\mathcal{L}_{Graph}$ is irrelevant to the predictions, which can also be viewed as $\eta_3=0$. We will elaborate on this in the revised manuscript.
>
> Weakness 3: In our study, we focus on leveraging the auxiliary critic domain to enhance user domain predictions. To avoid overcomplicating the probabilistic derivation process, we did not use a VAE in the critic embedding network. Note that $v_j^a$ is involved in the loss function $\mathcal{L}_{VAE}$, implicating $u_l^c $ in the VAE's ELBO for $u_i$. Employing two VAEs for $u_i$ and $ u_l^c $ would significantly complicate the VAE Loss due to their interdependence. Thus, we opted for a VAE solely in the user domain but not the auxiliary critic domain.
>
> As for your concern “why not use the (analogous) first term in equation of 5 in equation 3”, this is because equation 5 is for concurrently learning the latent critic-rating and critic-comment vectors while equation 3 is only for learning the User-Comment vectors. The first term in equation of 5 is for learning critic-rating vectors. As discussed in the paper, given the diverse nature of user comments and their potential limited correlation with item property evaluations [1,2,3] and significant correlation exists between critics' scores and their explanatory comments [1,2], we opt for independent learning for users while concurrent learning for critics. According to your comments, we will expand on the rationale behind the loss design in our revised version.
>
> Response to Questions:
>
> Question 1: We apologize for the typos. $\eta_s$ should be $\eta_1$, $\eta_c$ should be $\eta_2$, and $\eta_n$ should be $\eta_4$. These will be corrected in our revised paper.
>
> Question 2: Sorry for the confusion on the ablation study C1. It excludes review text information for both user and critic embedding networks. Specifically, we set the user-comment vector $w_i=0$, and the loss $\mathcal{L}_{MSE}$ is not applied.
>
> Additionally, the term $| y_{lj}^c-z_{lj}^c|  ^2$ is omitted in the loss $\mathcal{L}_{Multi}$. We also replace $v_j^a$ with the item-summary vector $v_j$. More details will be provided in the revised version.
>
> Question 3: Ablation study C2 involves replacing $v_j^a$ with the item-summary vector $v_j$.
>
> Question 4: Ablation study C3 involves not applying the loss $ \mathcal{L}_{Multi} $ and replacing $v_j^a$ with $v_j$.
>
> Additionally, all critic-item edges are omitted from the graph for the loss $\mathcal{L}_{Graph}$.
>
> Question 5: Yes. Ablation study C4 mean $\eta_4=0$.
>
> Owing to the constraints of space, we will provide the remaining responses in our subsequent reply.

---

> > ### Author Response · Authors · 2023-11-15
> >
> > Question 6: Thank you for the question on the computational cost. MCIR combines VAE (using MLPs) and GCN architectures that have good scalability with complexity linear to the number of users, critics, and items. MCIR mainly contains VAE (composed of MLPs) and GCN architecture (the complexity is as same as LightGCN). By comparison, BitGCF mainly contains GCN architecture while EMCDR and SSCDR mainly contains MLPs. First, from the training time perspective, MCIR requires approximately 10 seconds per epoch on the MC-Game dataset, compared to 3 seconds for EMCDR and SSCDR, and 30 seconds for BitGCF. Therefore, MCIR can compute efficiently while outperform all the baseline methods.
> > In terms of parameter count, MCIR is on par with these models, with the number of GCN parameters being $(N_u + N_w + N_v) d$ and MLP parameter complexity dependent on layer dimensions. The number of GCN parameters in BitGCF is similar to MCIR and the number of MLP parameters in EMCDR and SSCDR are similar to those in MCIR.
> > As for inference cost, MCIR, along with EMCDR and SSCDR, employs MLPs for efficient inference, whereas BitGCF's reliance on GCN layers results in a marginally higher cost.
> >
> > Question 7: Thank you for your suggestion. Actually, we have tried the concatenation treatment in Equation 7. Our experiments with concatenation did not yield significant performance differences, leading us to opt for an additive approach to maintain model simplicity.
> >
> > Question 8: Thank you for your suggestion. Indeed Equation 8 is quite simple and effective. We experimented with additional project layers but found no substantial improvement in performance, thus opting for the simpler model.
> >
> > Question 9: We thank you for your suggestion regarding dataset choice. Our focus was on cross-domain item-sharing recommendations, integrating user preferences with critic opinions. Existing datasets did not encompass critic data, prompting our choice of Metacritic. Our collected data will be publicly available after the paper is accepted.
> >
> > Again, it is truly grateful for your comments, we will carefully refine the paper according to your suggestions. Should you have any further questions or require additional clarification, please do not hesitate to contact us.
> >
> > [1] Santos T, Lemmerich F, Strohmaier M, et al. What's in a review: Discrepancies between expert and amateur reviews of video games on metacritic[J]. Proceedings of the ACM on human-computer interaction, 2019, 3(CSCW): 1-22.
> >
> > [2] Dillio R. Different Scores: Video Gamers' Use of Amateur and Professional Reviews[M]. Rochester Institute of Technology, 2013.
> >
> > [3] Parikh A A, Behnke C, Almanza B, et al. Comparative content analysis of professional, semi-professional, and user-generated restaurant reviews[J]. Journal of foodservice business research, 2017, 20(5): 497-511.

---

> > > ### Author Response · Authors · 2023-11-21
> > >
> > > Dear Reviewer 6fW2,
> > >
> > > I hope this message finds you in good health. First and foremost, I want to extend our sincere gratitude for the time and effort you have dedicated to reviewing our paper. Your insights are invaluable to us.
> > >
> > > We have carefully considered and addressed the comments and concerns you raised in your review. We would greatly appreciate it if you could share your thoughts on our rebuttal. We are particularly keen to understand your perspective after reviewing our explanations. Any further feedback or suggestions would be immensely helpful in guiding potential improvements to our paper.
> > >
> > > Thank you again for your valuable contribution to this process. We look forward to your response and are committed to making all necessary enhancements to our work.
> > >
> > > Best regards,

---

> > > > ### Comment · Reviewer_6fW2 · 2023-11-21
> > > >
> > > > Thank you for the clarifications. Please include them (and more) in the main draft. There are several parts in the pipeline, and it is hard for the first time reader to appreciate the value, need and intuition for the individual parts.
> > > >
> > > > For Question 2 above, do you also mean to set $w^c_l =0$ in $\mathcal L_{multi}$? These changes do make me more optimistic about the paper..

---

> > > > > ### Author Response · Authors · 2023-11-22
> > > > >
> > > > > Thank you very much for your response.
> > > > >
> > > > > Regarding Question 2, you are correct in your understanding. In the case of $\mathcal{L}_ {Multi}$, the term $w_l^c$ can effectively be considered as 0. This is because the term $| y_{lj}^c-z_{lj}^c|^2$ is omitted from the loss, and we replace $v_j^a$ with the item-summary vector $v_j$. As a result, $w_l^c$ does not participate in the learning process, and the term $| w_l^c|^2$ in $\mathcal{L}_{Multi}$ can be viewed as having no impact on the other embeddings.
> > > > >
> > > > > We will ensure to incorporate all the discussed details and clarifications in the final version of the paper. Your insights have been instrumental in enhancing the comprehensibility and overall quality of our work.
> > > > >
> > > > > We appreciate your time and effort in reviewing our paper. We hope that our responses have adequately addressed your concerns. If you find our explanations satisfactory, we kindly request that you consider raising the rating for our paper. However, if you still have any unresolved issues or require further clarification, please do not hesitate to let us know.

---

### Meta-Review · Area_Chair_ZHox · 2023-12-27

**Metareview:**

The authors study the problem of cross-domain recommendation systems where: (1) the items are common across 'domains', (2) domains are defined by the population of users (in this paper, critic reviewers vs. user reviews) and (3) there is a (ostensibly content-based) relationship between preferences expressed by the two disjoint populations. This is denoted as a "cross-domain item-sharing recommender system" (CIR) and the specific solution provided is based on defining multiple views within the domains (e.g., reviews vs. ratings), thus referred to as a "multi-view  cross-domain item-sharing recommender system" (MCIR). Once multiple-views are encoded/embedding, an attentive integration mechanism, graph embedding network, and cross-view contrastive learning is used to fine-tune the embeddings and harmonize them across multiple views. Experiments are conducted on datasets collected by the authors from metacritic regarding games, movies, and music -- showing notable empirical improvements over (modified) existing cross-domain recommendation methods with widely-used metrics (e.g., Recall@k, Precision@k, MAP@k, NDCG@k). Additional ablation studies are performed to estimate the relative performance contribution of different proposed architecture components.

Consensus strengths identified by reviewers regarding this submission (that I also concur with) include:
- Novelty of the problem: While cross-domain recommendation system works are common, the particular configuration proposed in this paper is unique and practical for specific settings.
- Clarity of writing: From a narrative writing perspective, this submission is well-structured and uses figures to better elucidate the methods and motivation (although aligning Figure 2 with notation in the paper could be improved).
- Empirical performance and setup: While the authors specifically focus on the 'critic vs. user' populations, they do create multiple datasets and show strong performance relative to the most closely related baselines. While the experimental setup wasn't trivial as this is a new problem, the reviewers felt the experiments were comprehensive and convincing.

Conversely, consensus limitations included:
- Empirical limitations: While partially addressed during rebuttal, there were many questions regarding evaluating the relative contribution of different architectural elements -- and sufficient discussion of these aspects of the algorithm.
Convincing motivation: While the setting is novel and the authors have identified a particular use-case, it isn't clear how impactful this setting would be (i.e., is it an esoteric problem)? I think this was reflected in some confusion about the specified setting (i.e., 'domains' are defined by their disjoint populations while the items are the same; domains are usually associated with disjoint items).
General writing issues: There were many technical clarifications that were partially addressed during rebuttal, but probably need another review to ensure consistency and clarity.

In my reading of the paper, I had the following primary concerns:
- I think the problem is a bit esoteric. Thus, while potentially important for RecSys, I think there will be limited impact across the broader community. Conversely, if the datasets are provided, I think there is room for further improvements on this specific setting *and* related settings.
- It isn't clear that the experiments are 'apples to apples'. Instead of focusing on many baselines (although I wouldn't throw this away), I would want to better understand how the existing algorithms were modified (after all, they weren't actually designed to solve this problem). This can probably be resolved with discussion, but I believe this is necessary.

Overall, the reviewers agreed that it was an interesting setting and the results were promising, but additional work is needed to make this a strong submission.

**Justification For Why Not Higher Score:**

The reviewers raised consistent and valid concerns regarding a convincing motivation, improved technical description of the method, and more discussion regarding the empirical results.

**Justification For Why Not Lower Score:**

N/A

---

### Decision · Program_Chairs · 2024-01-16

Reject